# Graph Aggregation Beyond Homophily Assumption

## a more meaningful way to model networks

## Abstract

Data aggregation on Homophily/Heterophily networks have caused lots of discussions. Existing solutions are all based on Homophily assumption that Heterophily edges are considered as noisy data and need to be eliminated. In this paper, we first conduct a case study to show data aggregation can not be affected by network types, but aggregation strategies. Graph Weighted Aggregation (GWA) method is proposed to perform aggregation with three attributes: (1) node features, (2) network topology and (3) label information. We also propose to use Riemannian manifold to model topological networks with Ricci Curvature as the force of influence between adjacent nodes. The three attributes together can formulate a strategy to aggregate neighboring nodes through message passing on Graph Neural Network (GNN). This methodology defines a more meaningful way to aggregate neighboring nodes with no regard to Homophily assumption. GWA algorithm outperforms the state-of-the-art algorithms on benchmark datasets.

## CCS Concepts

• **Computing methodologies** → **Neural networks**.

## Keywords

Graph Neural Networks, Homophily Assumption, Node Aggregation, Ricci Curvature

### ACM Reference Format:

. 2026. Graph Aggregation Beyond Homophily Assumption: a more meaningful way to model networks. In . ACM, New York, NY, USA, 7 pages. https://doi.org/10.1145/nnnnnnn.nnnnnnn

## 1 Introduction

In datasets where both node-level features and inter-sample associations (i.e., network/graph structures) are present, it is essential to understand how feature values contribute to classification and how topological relationships influence sample behavior. Feature sets typically inform categorization or label prediction, while graph topology reflects how a sample influences — or is influenced by — its neighbors. In classification tasks, incorporating problem-specific structural and feature-based patterns into the model can enhance predictive accuracy and interpretability. This gives rise to a key modeling challenge: how to systematically integrate both feature-driven and structure-driven information into a unified learning framework.

In many real-world datasets, similarities among samples give rise to graph structures, where edges are formed based on shared characteristics. However, such structural connections often emerge from multiple overlapping features, and it is rarely evident which of these features contribute primarily to edge formation versus those that inform classification. Most existing models do not explicitly disentangle these roles, treating all features as equally relevant to both connectivity and categorization. This ambiguity complicates the modeling process and can lead to suboptimal performance when structure and label information are not well aligned.

In graph-based models, particularly GNNs, the representation of each node is influenced not only by its own features but also by the features of its neighbors. This results in feature smoothing or aggregation, where a node's feature vector is updated based on a weighted combination of its neighbors' features. Intuitively, this reflects the principle that a node's identity is shaped by its local context — a notion often summarized as being defined by one's neighborhood. Formally, this process can be represented as a weighted average, where the contribution of each neighboring node is modulated by an associated influence or attention weight.

The adage "birds of a feather flock together" succinctly captures the principle of homophily, which posits that nodes in many real-world networks tend to connect to others with similar attributes or labels [16]. While homophily is a foundational assumption in numerous graph-based models, it often oversimplifies complex network structures. In reality, connectedness and categorization represent distinct phenomena: nodes from different categories can and frequently do establish connections. Such networks, where edges link nodes of differing labels, are referred to as heterophily networks, highlighting the limitations of homophily-based assumptions.

Existing approaches to this challenge primarily focus on aggregating information across varying levels of homophily. These levels encompass structural features such as 1-hop, 2-hop, up to n-hop connections, node-similarity metrics like K Nearest Neighbor (KNN) links, and domain-specific connections, including same-category edges. However, these methods typically rely on the homophily assumption and attempt to reshape the graph structure to approximate homophilous networks, often by diminishing the influence of heterophilous and random connections.

The challenge can also be reframed as uncertainty regarding how positive and negative labels are defined within the feature space; in many applications, the precise feature-label relationships remain unknown. Existing methods often assume that all features contribute solely to categorization and not to network connectivity. Moreover, they presuppose that both feature sets and topological properties align consistently with the labels and support categorization. Consequently, these approaches seek to diminish or eliminate edges connecting nodes with differing labels. However, such assumptions overlook critical complexities and may undermine modeling effectiveness.

Our approach begins by selecting multiple feature sets, including node attributes, topological properties, and label information. We model node features and network topology independently, then adjust neighbor influences by incorporating both their topological characteristics and label information. This aggregation strategy addresses the limitations of homophily-based methods by removing the homophily assumption and explicitly accommodating the coexistence of homophily, heterophily, and random connections within networks.

Our key contributions are summarized as follows:

- We conduct a case study simulating the graph aggregation process, demonstrating that aggregation functions can exert a greater influence on node representation than the underlying network types.
- We introduce a dual-model approach to address graph aggregation: one model leverages Ricci curvature to capture topological structure, while the other employs Graph Neural Networks to represent node features. These models are integrated via weighted aggregation implemented through message passing in the GNN framework.
- We propose a modification to the Forman-Ricci curvature formula to achieve stability, transforming the existing sectional local Forman-Ricci curvature into a stabilized global version. We prove that this modified curvature holds when the network reaches a minimum energy state.
- Through extensive experiments on eight benchmark datasets, we demonstrate that our method outperforms current state-of- the-art approaches.

## 2 Related Work

Existing approaches to address the homophily and heterophily issues in graph learning can be broadly categorized into three groups: node selection, weighted aggregation, and label propagation.

### 2.1 Node Selection

SparseGAD [8] categorizes neighboring nodes into three groups: homophilic, heterophilic, and irrelevant, utilizing only homophilic neighbors for aggregation. This method was evaluated on three datasets, achieving accuracies of 65.82% on YelpChi, 89.17% on Amazon, and 5.98% on Reddit. H2GCN [31] proposes improvements at multiple levels to address heterophily, including 1-hop, 2-hop, and combined intermediate representation settings. The method assumes that higher homophily ratios correlate with improved performance. Similarly, ASP [6] constructs three views—original, attribute, and global structure—and integrates them to mitigate heterophily challenges.

### 2.2 Weighted Aggregation

Personalized Propagation of Neural Predictions (PPNP) and its efficient variant APPNP [9] leverage the connection between Graph Convolutional Networks (GCN) and PageRank to develop an enhanced propagation mechanism resembling a random walk. Evaluations on CiteSeer yielded accuracies of 75.83% (PPNP) and 75.73% (APPNP); on Cora, 85.29% and 85.09% respectively; and on PubMed, APPNP achieved 79.73% while PPNP was inapplicable. Generalized PageRank (GPR) [7] optimizes node features using learned PageRank weights, reporting accuracies of 79.51% on Cora, 67.63% on CiteSeer, 85.00% on PubMed, 67.48% on Chameleon, 49.93% on Squirrel, 92.92% on Texas, and 91.36% on Cornell.

### 2.3 Label Propagation

Label Propagation (LPA) and Graph Convolutional Networks (GCN) share a conceptual similarity in propagating labels and features, respectively, via smoothing operations [29]. In LPA, a node's label is updated as a weighted average of its neighbors' labels, while in GCN, node representations are updated through a weighted average of neighboring node features. This method demonstrated competitive performance with accuracies of 88.5% on Cora, 78.7% on CiteSeer, 87.8% on PubMed, 94.8% on Coauthor-CS, and 96.9% on Coauthor-Phy.

### 2.4 Curvature-enhanced Network Analysis

Ricci curvature has been discretized for application to networks. Ollivier-Ricci curvature [17], grounded in Riemannian geometry, characterizes diffusion and stochastic properties but is computationally intensive for large-scale networks. Forman-Ricci curvature [8], based on topological considerations, offers computational efficiency and relates to the classical Laplace operator, serving as an abstract analog to the Bochner-Weitzenböck formula from differential geometry. Additionally, Menger-Ricci and Haantjes-Ricci curvatures [25], defined via spherical triangulation of networks, share the topological foundation of Forman-Ricci curvature and are likewise computationally tractable for practical use.

## 3 Motivating Observations

This case study is to test the impact of the topological aggregation on different networks, such as Homophily, Heterophily and Random networks.

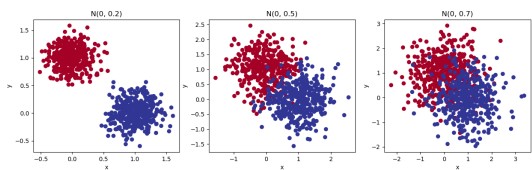

**Figure 1: Visualization of Sample sets in Gaussian Distribution with Two-Class Centers at [0, 1] and [1, 0]**

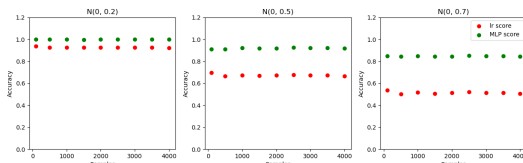

**Figure 2: Classification Performance on Three Sample Sets**

We design a two-step process to present the effect of different interactions between topological properties and feature sets.

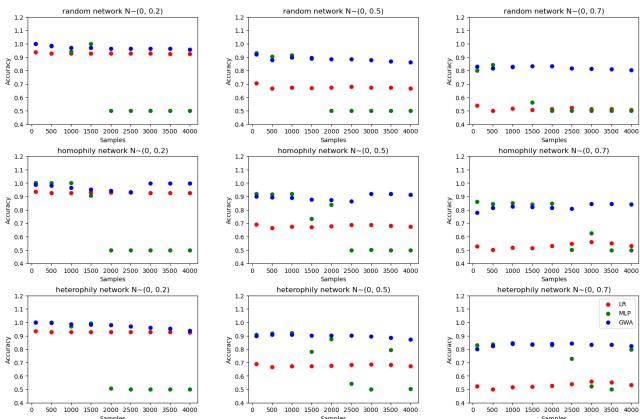

**Figure 4: Visualization of Classification Performance on Different Types of Networks with Different Aggregation Methods**

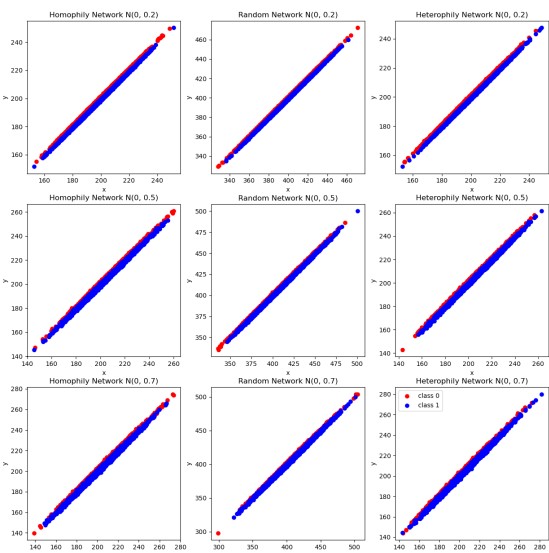

**(a) Visualization of 4000 Vertices After 1-Hop Aggregation**

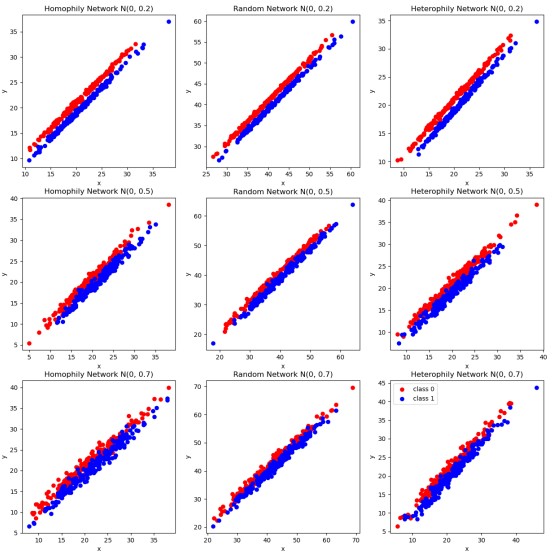

**(b) Visualization of 400 Vertices After 1-Hop Aggregation**

**Figure 3: Visualization of Three Types of Networks after Aggregating Neighbors**

- Step 1: we generate two-class sample sets around the centers (0, 1) and (1, 0), respectively, with Gaussian distribution N(0, 0.2), N(0, 0.5), and N(0, 0.7), as shown in Figure 1, and evaluate data quality with both Linear Regression (LR) and Multi-Layer Propagation (MLP) algorithms, as shown in Figure 2.
- Step 2: we generate three different networks, such as Homophily networks (which only connect samples within the same classes), Heterophily networks (which only connect samples in different classes), Random networks (which randomly connect samples, no matter which class they belong to). In Figure 3, we aggregate neighboring samples in different networks through addition and visualize the results. In Figure 4, we aggregate neighboring samples through both addition and weighted addition by using node attributes, topology information and label information, and evaluate the data quality through node classification.

In Figure 2, when the standard deviation increases from 0.2 to 0.7, data points in the two classes become more overlapped. Meanwhile, the classification performance of both Linear Regression (LR) and Multi-Layer Propagation (MLP) decreases, because, the overlapping makes it more and more difficult to classify the two classes.

In Figure 3, when the aggregation through addition is conducted on the three different types of networks, the values of the vertices are changed in the same way that the values are converged to the line rotating 45° from x axis in Quadrant I. When the number of samples increases from 400 to 4000, the values are converged faster. Based on the visualization, the aggregation strategy of data addition can not differentiate the characteristics of the two classes.

In Figure 4, LR and MLP use the same aggregation function and performs almost the same, but GWA uses a different aggregation function which considers the combine effect of node information, topological information and the label information. GWA performs better than LR and MLP on different network types on all the sample sets. The accuracy of LR and MLP decreases from 90% to 50%, when

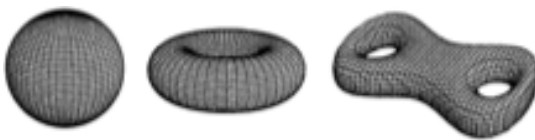

(a) Riemannian Surfaces of Different Genera [4]

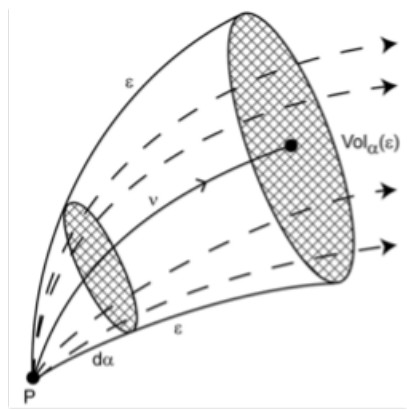

(b) Geometric Interpretation of Ricci Tensor [26]

Figure 5: Ricci Curvature on Riemannian Manifold

standard deviation increases from 0.2 to 0.7; the accuracy of GWA is between 80% to 90% on all the datasets.

Based on this observation in Figure 3 and Figure 4, we can draw the conclusion that, for different network types, the effect of data aggregation is related to the aggregation strategy. Aggregation function formulates the interactions between neighboring nodes through node information, topological information and label information. It determines the contribute of individual features to node classification.

## 4 Graph Weighted Aggregation (GWA)

A manifold defines a space that may be curved and have a complicated topology, but in local regions looks just like Euclidian space $R^n$ [5]. Every patch of the manifold must have the dimensionality n of the Euclidean space. If all the signs in the metric of the manifold are positive, the space is called Riemannian Manifold; if there is a single minus sign, it is called pseudo-Riemannian Manifold [5].

The Riemannian tensor has four indices [5]. At times, it is useful to express a tensor as a sum of various pieces that are individually easier to handle and may have direct physical interpretations. Ricci tensor is formed by taking the contraction of the Riemann tensor. For the curvature tensor formed from an arbitrary connection, there are a number of independent contractions to take. The Ricci tensor associated with the Christoffel connection is automatically symmetric as a consequence of the symmetries of the Riemann tensor. The trace of the Ricci tensor is the Ricci scalar (or curvature scalar).

As shown in Figure 5, a point p in a manifold M (shown in Figure 5(a)) can be denoted as the vectors at p in the Tangent Space which is merely an abstract vector space associated with each point p in a

### Table 1: Descriptions of Symbols

| Symbol | Description |
|--------|-------------|
| $F(v)$ | Forman-Ricci curvature for node v |
| $w_e$ | Weight associated with edge e |
| $e_{v_i}$ | The edge containing $v_i$ |
| $\lambda_v$ | The eigenvalue of node v |

manifold M. Tangent space is the set of all vectors at a single point in spacetime. It is merely an abstract vector space associated with each point p in a manifold M. After mapping Riemannian manifold to Tangent Space, we can use Ricci tensor to compute the value of the Ricci curvature. Ricci tensor encodes all the essential properties of a Riemannian metric [5], as shown in Figure 5(b).

To apply Riemannian geometry to networks, there are several discrete notions of Ricci curvature, such as Ollivier-Ricci curvature [17] [18] [19] [20] Forman-Ricci curvature [8], Menger-Ricci curvature [25], Haanjes- Ricci curvature [25], etc. Most of the notions are sectional local discretization of Ricci curvature, each capture different geometric properties and has different drawbacks because of the lacking of the smoothness in network structure [25]. In other words, in the network context, it is impossible to find the best curvature because each notion can only code a subset of properties for a certain task in a specific type of networks. In this work, we choose Forman-Ricci curvature as the foundation.

Another challenge is the higher-order interactions in network dynamics. Different networks [3] [1] [2] encodes the dynamics and the evolutions through the interactions between cells in different settings, which can be represented through graph properties such as degree centrality, clustering coefficient, eigenvector centrality, etc. The driving force of the interaction between vertices is Ricci curvature which is flowing, merging and distributing through network paths to all directions. However, for computation purpose, we need to capture the stable state of the energy in the networks for a particular moment so that we can systematically model the properties of the networks for that moment.

Eigenvector centrality can be used to measure the influence of the nodes in networks [11]. In adjacency matrix, the more connections a node has, the larger the eigenvalue. Different from degree centrality, eigenvector centrality is computed through several iterations. With the computation progresses, nodes with more connections start gaining more importance until the values are stabilized when the graphs or networks reach the lowest energy.

### 4.1 Modification of Forman-Ricci Curvature

**Notations:** We list the descriptions of the symbols in Table 1.

Forman-Ricci curvature in the 1-dimensional case, such as graphs or networks, can be represented in the following formula [24]:

$$F(a) = w_a\left(\frac{w_{v_1}}{w_a} + \frac{w_{v_2}}{w_a} - \sum_{e_{v_1} \sim a, e_{v_2} \sim a}\left(\frac{w_{v_1}}{\sqrt{w_a w_{e_{v_1}}}} + \frac{w_{v_2}}{\sqrt{w_a w_{e_{v_2}}}}\right)\right) \quad (1)$$

in which a is the edge under consideration, $v_1$ and $v_2$ are two vertices, $e_{v_1}$ and $e_{v_2}$ are the set of edges containing $v_1$ and $v_2$, respectively, after excluding the edge e. $w_a$ is the weight associated

with edge a, $w_e$ is the weight associated with edge e, and $w_v$ is the weight associated with vertex v, such as $w_{v_1}$ is the weight for vertex $v_1$ and $w_{v_2}$ is the weight for vertex $v_2$.

For $w_a = w_e = w_v = 1, F(a) = 4 - \sum_{v \sim a} deg(v)$.

THEOREM 1 (EDGE CURVATURE). *When graphs or networks become stabilized and reach the lowest energy, the following formula holds.*

$$F(a) = w_a(\lambda_{v_1} + \lambda_{v_2}) \tag{2}$$

*in which $w_a$ is the edge weight, $v_1$ and $v_2$ are the two ends of the edge, and $\lambda_{v_1}$ and $\lambda_{v_2}$ are the eigenvalues of $v_1$ and $v_2$, respectively, when the graphs or networks are converged.*

PROOF. For $w_v = \lambda_v$, in which $e \in E(G)$ and $v \in V(G)$, $\lambda_v$ is the weight associated with vertex v when graphs/networks are stabilized - the eigenvalue of node v, we have

$$F(a) = w_a * (\lambda_{v_1} + \lambda_{v_2}) - w_a * \sum_{e_{v_1} \sim a, e_{v_2} \sim a}(\frac{\lambda_{v_1}}{\sqrt{w_a * w_{e_{v_1}}}} + \frac{\lambda_{v_2}}{\sqrt{w_a * w_{e_{v_2}}}})$$

$$= w_a*(\lambda_{v_1} - \sum_{e_{v_1} \sim a}(\frac{\lambda_{v_1}}{\sqrt{w_a * w_{e_{v_1}}}})) + w_a*(\lambda_{v_2} - \sum_{e_{v_2} \sim a} \frac{\lambda_{v_2}}{\sqrt{w_a * w_{e_{v_2}}}})$$

Let $g_a(v) = w_a * (\lambda_{N_a(v)} - \sum_{e_{N_a(v)} \sim a}(\frac{\lambda_{N_a(v)}}{\sqrt{w_a * w_{e_{N_a(v)}}}}))$. It represents the total forces v can receive from Neighboring vertices through edge a, $N_a(v_1) = v_2$ and $N_a(v_2) = v_1$, then we have $F(a) = g_a(v_1) + g_a(v_2)$

For $g_a(v) = w_a * (\lambda_{N_a(v)} - \sum_{e_{N_a(v)} \sim a}(\frac{\lambda_{N_a(v)}}{\sqrt{w_e * w_{e_{N_a(v)}}}}))$,

$\sum_{e_{N_a(v)} \sim a}(\frac{\lambda_{N_a(v)}}{\sqrt{w_e * w_{e_{N_a(v)}}}}) = 0$. Because, when $w_v = \lambda_v$, adjacency matrix is stabilized so that energy propagation is not under consideration, the forces passing from higher-order vertices can be ignored.

We have $g_a(v) = w_a * \lambda_{N_a(v)}$, and then, $F(a) = w_a * \lambda_{N_a(v_2)} + w_a * \lambda_{N_a(v_1)} = w_a(\lambda_{v_1} + \lambda_{v_2})$

We can conclude $F(a) = w_a(\lambda_{v_1} + \lambda_{v_2})$ holds, when graphs or networks reach the lowest energy. □

THEOREM 2 (NODE CURVATURE). *When graphs or networks become stabilized and reach the lowest energy, the following formula holds.*

$$F(v_i) = \sum_{v_j \sim N(v_i)} w_{v_i v_j} * \lambda_{v_j} \tag{3}$$

*in which $w_{v_i v_j}$ is the edge weight, $v_i$ and $v_j$ are the two ends of the edge, and $\lambda_{v_i}$ and $\lambda_{v_j}$ are the eigenvalues of $v_i$ and $v_j$, respectively.*

PROOF. $F(v_i) = \sum_{v_j \sim N(v_i)} F(v_j)$. Based on Theorem 4.1, we have $F(v_i) = \sum_{e_{v_i}} F(e_{v_i}) - w_{e_{v_i}} * \lambda_{v_i} = \sum_{v_j \sim N(v_i)} w_{v_i v_j} * \lambda_{v_j}$

Especially, when $w_{v_i v_j} = 1, F(v_i) = \sum_{v_j \sim N(v_i)} \lambda_{v_j}$ □

## 4.2 Message Passing

The classic GNN through message passing is the fundamental architecture to aggregate node information through network topology. GWA algorithm improves GNN by introducing the momentum of the network that the initial information in the network is considered as local information and the global information of the network can be obtained by aggregating neighboring information, until the network reaches the minimum energy.

**Table 2: Data Description**

| Dataset | Nodes | Edges | Features | Classes |
|---|---|---|---|---|
| Cora | 2708 | 5429 | 1433 | 7 |
| PubMed | 19717 | 44338 | 500 | 3 |
| CiteSeer | 3327 | 4732 | 3703 | 6 |
| Cornell | 183 | 298 | 1703 | 5 |
| Wisconsin | 251 | 515 | 1703 | 5 |
| Texas | 183 | 325 | 1703 | 5 |
| Chameleon | 2277 | 36101 | 2325 | 5 |
| Squirrel | 5201 | 217073 | 2089 | 5 |

During model training, the network continuously aggregates neighboring information and gradually becomes stable, until the network information is converged. As shown in Formula 4, M is the weighted adjacency matrix which is generated through message passing, W is the trainable weight matrix. $M^p$ is obtained by propagating information from nodes that are p-hop away in the network. C indicates the label information which is the class of the node, E indicates eigenvalues of the nodes, A is the adjacency matrix and X is the node information. Note that p is not allowed to be 0. However, when p is 0, GWA is turned into a one-layer MLP.

$$Z = M^p * X * W \tag{4}$$

$$M = C * E * A \tag{5}$$

When the network becomes stable, one-time aggregation through the topology structure can not make any changes. Because of this, during prediction, as shown in Formula 6, we disregard the structural information and only use node information as the input.

$$Z = X * W \tag{6}$$

## 5 Experiments

### 5.1 Experimental Setup

**Datasets.** We evaluated the performance of our methodology by using eight real world datasets (as shown in Table 2) in which Cora [15], CiteSeer [10] and PubMed [27] data are citation networks, Chameleon and Squirrel data [23] are Wikipedia networks, and Cornell, Wisconsin and Texas data [1] are webpage networks.

**Baselines.** We compared our methodology with ten baselines: GCN [14], GAT [28], GraphSAGE [12], JK-Net [30], SSP [22], Geom-GCN [21], GCN-LPA [29], U-GCN [13].

**Parameter Settings.** For all the methods, we use Cross Entropy loss function, Adam optimizer with learning rate 0.01 and weight decay 5e-4, and dropout rate 0.5. We used the same splits for training, testing and validation sets. We reported 10 times' averages in Table 2.

### 5.2 Node Classification

We evaluated the model performance with accuracy and compared the performance of our methodology with other node classification algorithms, as shown in Table 3. On all the 8 benchmark datasets,

---

[1] https://www.cs.cmu.edu/afs/cs.cmu.edu/project/theo-20/www/data/

GWA outperformed other algorithms. On Cora, GWA improved the state-of-the-art accuracy by 18.48%. On PubMed, GWA improve the state-of-the-art accuracy by 4.49%. On Citeseer, GWA improved the state-of-the-art accuracy by 34.50%. On Cornell, GWA improved the state-of-the-art accuracy by 37.84%. On Chameleon, GWA improved the state-of-the-art accuracy by 69.59%. On Squirrel, GWA improved the state-of-the-art accuracy by 116.98%. On Wisconsin, GWA improved the state-of-the-art accuracy by 37.95%. On Texas, GWA improved the state-of-the-art accuracy by 37.14%. We also reported how much the accuracy was improved in comparison with the state-of-the-art performance, as shown in Table 4.

Because MLP does not utilize the network topology to model the data, MLP as the benchmark solution for node classification, was also compared in the improvement of accuracy to the state-of-the-art performance. MLP on the 8 benchmark datasets underperformed the state-of-the-art solutions. On Cora, MLP underperformed the state-of-the-art algorithm by 2.46 %. On PubMed, MLP underperformed the state-of-the-art algorithm by 3.19 %. On Citeseer, MLP underperformed the state-of-the-art algorithm by 8.56 %. On Cornell, MLP underperformed the state-of-the-art algorithm by 5.56 %. On Chameleon, MLP underperformed the state-of-the-art algorithm by 23.52 %. On Squirrel, MLP underperformed the state-of-the-art algorithm by 14.39 %. On Wisconsin, MLP underperformed the state-of-the-art algorithm by 8.14 %. On Texas, MLP underperformed the state-of-the-art algorithm by 8.13 %. In average, MLP underperformed the state-of-the-art algorithm by 9.23 %. This difference was caused by the utilization of the topology structure in data modeling.

## 6  Limitation

This model can be applied to the networks on which node features can be influenced by the context, such as text, relationships, etc. When there is information exchange through the information channels, certain node features can be updated through the information exchange. Some node features can not be aggregated, such as date of birth, etc.

## 7  Conclusion

We proposed a weighted aggregation algorithm - GWA. We extended Forman-Ricci curvature theory and use graph properties to compute Forman-Ricci curvature in social networks. We tested GWA on synthetic data and eight real-world data sets to prove that node information, the label information and network topology can be used to define an aggregation strategy for social networks. In comparison with the state-of-the-art homophily-heterophily solutions, GWA outperforms the state-of-the-art solutions on the eight benchmark data sets.

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

**Table 3: Model Accuracy (Percentage) on 8 Datasets (The best performance is bolded in red and the second one in black)**

| Methods | Cora | PubMed | Citeseer | Cornell | Chameleon | Squirrel | Wisconsin | Texas |
|---|---|---|---|---|---|---|---|---|
| GCN | 82.93 | 83.29 | 73.12 | 46.51 | 52.32 | 33.10 | 47.73 | 52.71 |
| GAT | 83.13 | 84.42 | 72.04 | 48.06 | 51.38 | 32.27 | 46.59 | 49.61 |
| SSP | 81.08 | 79.50 | 71.13 | 55.04 | 21.87 | 19.72 | 49.37 | 55.04 |
| JK-Net | 81.27 | **86.15** | 71.74 | 52.71 | 53.95 | 33.51 | 48.30 | 51.94 |
| Graph-Sage | 82.20 | 83.03 | 71.41 | 53.49 | 42.29 | 26.89 | 56.82 | 53.49 |
| Geom-GCN | 74.27 | 83.49 | 73.79 | 54.26 | 38.66 | 32.22 | 53.41 | 64.34 |
| GCN-LPA | 82.33 | 85.83 | 72.29 | 49.61 | 52.69 | 33.48 | 50.57 | 48.84 |
| U-GCN | **84.00** | 85.22 | **74.08** | **69.77** | **54.07** | **34.39** | **69.89** | **71.72** |
| MLP | 63.33 | 83.08 | 67.74 | 65.89 | 41.35 | 29.44 | 64.20 | 65.89 |
| GWA | **99.52** | **89.70** | **99.64** | **96.17** | **91.70** | **74.62** | **96.41** | **98.36** |

**Table 4: Improvement of Benchmark and GWA Models in Comparison with the State-of-the-art Accuracy (Percentage)**

| Methods | Cora | PubMed | Citeseer | Cornell | Chameleon | Squirrel | Wisconsin | Texas |
|---|---|---|---|---|---|---|---|---|
| MLP | -2.07 | -2.75 | -6.34 | -3.88 | -12.72 | -4.95 | -5.69 | -5.83 |
| GWA | 15.52 | 3.87 | 25.56 | 26.40 | 37.63 | 40.23 | 26.52 | 26.64 |

