# OpenReview forum: "Graph Aggregation Beyond Homophily Assumption - a more meaningful way to model networks"
_ACM.org/TheWebConf/2026/Workshop/TIME — TIME 2026 Poster_

### Official Review · Reviewer_SPGb · 2026-01-03
**This paper proposes Graph Weighted Aggregation (GWA). Experiments on citation datasets show improvements over baselines, but the theoretical contributions are underdeveloped and experimental rigor is limited.**

**Rating:** 6
**Confidence:** 3

**Review:**

### Evaluation of Quality, Clarity, Originality, and Significance

This paper proposes **Graph Weighted Aggregation (GWA)**, a method that leverages **Ricci curvature** to guide aggregation in Graph Neural Networks (GNNs), integrating node features and label information. Overall, the paper is **clearly written and presents an interesting idea** that challenges the standard homophily assumption in GNNs. While the theoretical foundations are **underdeveloped** and experiments lack some rigor, the work demonstrates **consistent empirical improvements** and is suitable as a **preliminary methods contribution for a workshop**.

### Strengths

**Novel Use of Ricci Curvature**
Applying geometric concepts from Riemannian geometry to model node influence provides a **creative perspective** beyond traditional GNN aggregation.

**Challenging Homophily Assumptions**
The case study demonstrates that **aggregation strategy matters more than network type**, encouraging exploration of heterophily-aware methods.

**Empirical Improvements**
Results on multiple citation datasets show **consistent gains over baselines**, indicating practical potential for improved GNN performance.

**Clear Motivation**
The paper articulates limitations of homophily-based aggregation and the need for methods that do not assume edge similarity.

### Weaknesses (Evaluation-Focused)

**Underdeveloped Theory**
Key concepts such as **“minimum energy”** are **not rigorously defined**, and proofs connecting Ricci curvature to eigenvalue-based aggregation are **incomplete**. The relationship to spectral GNN methods is also underexplored.

**Limited Experimental Rigor**
No ablation studies are provided to quantify the contribution of curvature, labels, or their combination. **Statistical significance is untested**, hyperparameter details are sparse, and toy case studies may not generalize.

**Missing Comparisons**
Recent heterophily-aware GNNs (e.g., **H2GCN, LINKX**) and spectral methods are **not included** in experimental comparisons.

**Scalability Concerns**
Eigenvalue computations are **computationally expensive** for large graphs, and the discrepancy between training and inference procedures is **unclear**.

**Clarity Gaps**
Algorithmic steps and matrix constructions (curvature, label, adjacency) are **not fully described**, which may hinder reproducibility.

### Minor Observations

- Some figures could be **simplified for clarity**.
- Dataset statistics and experimental setup are sometimes **confused in tables**.
- References in figures and captions should be **checked for accuracy**.

### Questions for the Authors

1. Can you provide **ablation studies** to show the contribution of each component (curvature, label, adjacency)?
2. How is the **label matrix constructed**?
3. Why is structural information discarded at inference, and does this affect performance?
4. Can comparisons to **recent heterophily-aware or spectral GNNs** be added?
5. How feasible is GWA for **large-scale graphs** with expensive eigenvalue computations?

### Suggestions for Improvement

- Clarify theoretical foundations, including **“minimum energy”** and eigenvalue convergence claims.
- Include **ablation studies** and statistical testing for robust evaluation.
- Compare against **recent heterophily and spectral GNNs**.
- Add **pseudocode** and define all matrices explicitly for reproducibility.
- Discuss **scalability strategies** for large graphs.

### Overall Assessment

This is a **promising methods paper** that introduces a novel idea of using **Ricci curvature in GNN aggregation**. While **theoretical justification is underdeveloped** and experimental evaluation could be stronger, the paper presents **consistent empirical improvements** and aligns well with a **preliminary workshop contribution**. With minor clarifications and additional experiments, the work could become a **valuable reference for heterophily-aware GNN research**.

---

### Official Review · Reviewer_HbqN · 2026-01-03
**The paper argues that graph aggregation performance depends more on aggregation strategies than on whether a network is homophilous or heterophilous. It proposes a Graph Weighted Aggregation (GWA) framework that combines node features, topology, and label information, using a Riemannian manifold with Ricci curvature to guide message passing via energy minimization. Experiments show that the approach outperforms state-of-the-art methods on benchmark datasets.**

**Rating:** 4
**Confidence:** 3

**Review:**

### Strengths
1. The paper addresses an important and active debate in graph learning ,that is, how to handle heterophily, making the contribution highly relevant to current GNN research.
2. By reframing the problem around aggregation strategies rather than network type, the work provides a refreshing and potentially unifying perspective for both homophilous and heterophilous graphs.
3. The integration of node features, topology, and label information into a single aggregation mechanism is conceptually appealing and more expressive than feature-only or topology-only approaches.
4. Modeling graphs on a Riemannian manifold and using Ricci curvature to guide message passing is novel and theoretically grounded, adding depth beyond heuristic aggregation schemes.
5. The reported improvements over state-of-the-art methods on benchmark datasets suggest the method is competitive and practically effective.

### Weakness
1. The paper has a formatting problem.
2. Writing and presentation should be improved.
3. Some figures lack are not cited ,for example in Fig.5 a
4. Computing Ricci curvature and operating on manifolds may introduce nontrivial computational costs. Runtime complexity and scalability to large graphs are not clearly evaluated.

---

### Official Review · Reviewer_pt1Y · 2026-01-04
**Promising concept, but flawed theory and unclear evaluation**

**Rating:** 4
**Confidence:** 3

**Review:**

## Originality and significance
**Strengths**

* The attempt to incorporate discrete curvature as a principled, geometry-inspired driver of message passing is interesting and taps into a growing line of geometric learning research.

**Weaknesses**

* GWA’s use of label information means it fundamentally requires labels to propagate during training. This limits its applicability to scenarios where a decent number of labeled nodes are present and connected. For scenarios with very few labels, the method might not work or would revert to a normal GNN

## Technical content and experimental validation
**Strengths**

* The paper evaluates GWA on 8 diverse benchmark datasets

**Weaknesses**

* The core theoretical claims (Theorem 4.1/4.2) appear mathematically unsound or insufficiently justified. The paper defines $\lambda_v$ as a “per-node eigenvalue” and uses it in curvature formulations. However, in standard spectral graph theory, eigenvalues are global properties of matrices (e.g. adjacency matrices), not attributes of individual nodes. What nodes have are eigenvector components (e.g., eigenvector centrality), not eigenvalues. The paper appears to conflate these concepts, which undermines the mathematical rigor of the formulation.

* The manuscript repeatedly refers to stability/lowest energy without defining an energy function, proving convergence, or connecting it to standard graph-dynamical systems or spectral optimization. This is a major gap. Authors should formalize the mathematics with a clear energy-based framework

* The reported results - 99.52% accuracy on Cora and 99.64% on Citeseer - are highly atypical. Such near-perfect scores suggest the possibility of methodological issues, including flawed or non-standard data splits, label leakage, or evaluation on mixed edges or labels. Authors should provide precise protocol details (splits, ensure strict label masking and resolve any leakage, reporting results in terms of mean $\pm$ standard deviation) and a public codebase to verify reproducibility

* The study lacks ablation experiments to isolate the contribution of each proposed component. This omission makes it hard to pinpoint which aspect of GWA is driving the performance gains, leaving a gap in fully understanding the method’s effectiveness

## Clarity
**Strengths**

* For the most part, the writing is clear and easy to follow

**Weaknesses**

* The paper didn't follow the double-column format required by the conference

---

### Author Rebuttal · Authors · 2026-01-12

We sincerely thank the reviewers and Program Chairs for carefully reading of our manuscript. We are particularly grateful for receiving appreciation on the novelty and the valuation of our research. We believe that the clarification of certain technical details can help us reach common understandings about this research. Below, we respond to each comment point by point and describe the corresponding revisions.


# Response to Reviewer pt1Y.


## Originality and Significance
### Strength

curvature is not the driver of the message passing, but the force between neighboring vertices which can be used to model the entire network. Message passing is the mechanism to update node information with neighboring nodes in the Graph Neural Network (GNN) architecture. The entire architecture was not changed in this research. We provided a formula to compute node information which is formula (5).

The novelty of this research is in data modeling. Network modeling has been generally discussed in Physics, especially in Astronomy. In the space, all the stars are individual nodes and the forces between the stars can be considered as edges. The nodes and the edges together form Space Network. The Space Network is not only constantly spinning but also growing and evolving. For many years, physicists have developed fundamental theories about the time, the space and everything in between. When we borrow those theories to interpret society networks here on Earth, all the behaviors on the social networks become understandable. It is just the beginning. I believe, later on, we will borrow more theories from Physics to model problems in social networks.

If we simply use eigenvalues as weights to aggregate neighboring nodes, this idea seems short of theoretical foundation. However, if we consider social networks are similar to Space Network, the forces between nodes can be formulated by curvatures and the curvatures can be computed based on network properties. This way of modeling a network and network related problems seems more scientific and more convincing.

### Weaknesses
There are several applications in GNN. For node classification, in the training set, all the nodes need to be labelled and the number of labelled nodes needs to be big enough to train a model.

## Technical content and experimental validation

### Weaknesses

For each node, the eigenvalue is global information which only needs to be computed once. Forman-Ricci curvature is related to edge weight and vertex weight and can be computed with Formula (1). However, when we extend Forman-Ricci curvature to graph space, we need to use graph properties to interpret Forman-Ricci curvature theory. Theorem 4.1 formulates the Forman-Ricci curvature on the edge with eigenvalues. Theorem 4.2 formulates the Forman-Ricci curvature on the vertices with eigenvalues.

When training GNN through propagation, we constantly check the loss to see if the graph is converged enough. During training, there is information exchange through information channels. The more iterations the model goes through, the further hops the graph propagates. The graph is considered to be stable and to have the lowest energy when the training loss is small enough (or the training process triggers early stopping)

When label information is considered in the propagation [24], it can trigger a big boost in model performance. Good model performance can be considered as a virtue. In section 5.1 Parameter settings, it says we used the same splits for training, testing and validation sets. When using the benchmark datasets, in order to compare the results with other people's work, we need to do the same preprocessing. It is not proper to manipulate the benchmark datasets.

As mentioned in the title, the proposed methodology is to define a meaning way to model networks. We defined a strategy to formulate the graph aggregation with all three components together. This strategy was proved with experimental results.

## Clarity
### Weaknesses

This error was fixed. I changed the documentclass from [manuscript, review] to [sigconf, review]

[24] Hongwei Wang and Jure Leskovec. 2020. Unifying graph convolutional neural networks and label propagation. arXivpreprint. arXiv:2002.06755 (2020).



# Response to Reviewer HbqN.

## Weaknesses:
1. the formatting issue was fixed by changing documentclass from [manuscript, review] to [sigconf, review] in Latex.
2. My team has been polishing this paper for a while.
3. the citation was fixed.
4. Eigenvector Centrality is one of the graph properties which only need to be computed once.

# Response to Reviewer SPGb.

## The theoretical foundation are underdeveloped:

This research was built on top of Space Network and borrowed theories, such as Riemannian Manifold, Riemannian Geometry, Ricci curvature, etc., from Physics, especially Astronomy. In Space Network, because of heat, wind, gravity, and random crashes, the interaction between the nodes are much more complicated than that on social networks. However, for many years, physicists have developed many theories to model the forces in Space Network. It is the beginning to find the similarity between Space Network and social networks and to use existing theories in Physics to model social networks. However, a few people have been doing this and have been getting good results as well.

Other than the aggregation strategy we developed, we also figured out how to use graph properties to compute Forman-Ricci curvature in social network, which can be considered as an extension of Forman-Ricci curvature provided in Theorem 4.1 for edge curvature computation and Theorem 4.2 for node curvature computation.

The reason we used a case study to demonstrate the motivation was because the model we worked on and  the theories we borrowed were too complicated. A case study with 200 to 4000 random samples can construct simple network prototypes and demonstrate a general issue in networks.

## underdeveloped theory:
The stability of a graph model can be measured by the loss. When the loss converges to the minimum value, we can say the graph model is stable. The graph stability, the minimum energy of the graph, and the minimum loss of the graph are the same thing.

The two proofs are to compute the values of the curvatures on edges and vertices, respectively. The first proof starts with the formular of Forman-Ricci curvature and ends up with an extended Forman-Ricci formular with graph properties, such as eigenvalues,  as variables.  Also, in section 4.2 Message Passing, equation (4) and (5) indicates how to use the label information, node information, such as node features and node eigenvalues, and adjacency matrix to perform message passing and to eventually model outputs Z. The backbone algorithm for this research is GNN.


## Limited Experimental Rigor.

1. The aggregation strategy is to use node information, label information and structural information together to train a representation for node classification. The three components can not be separated. We compared the performance of our solution with the state-of-the-art solutions and the MLP as the benchmark solution.
2. The hypothesis was proved by the experimental results.
3. We reported all the hyperparameters in section 5.1 Parameter Settings.
4. In the case study, data points with Gaussian Distribution and edges are randomly generated to create general networks for the three types.

## Missing Comparisons:

In section 2 Related Work, we put the state-of-the-art research for Homophily-heterophily issue into several categories: node selection, weighted aggregation, label propagation. Especially, we cited H2GCN in related work section. In section 5 Experiments, we compared out algorithm with the state-of-the-art research cited in Section 2 in all the categories. We reviewed H2GCN and LINKX algorithms. Our work and selected algorithms outperformed H2GCN and LINKS.

## Scalability Concerns

Given a graph, it has several topological properties. Eigen centrality is one of the properties and is global information for the entire graph, so that we only need to compute eigenvalues once.

## Clarity Gaps

The equation (5) indicates how to perform matrix computation. Our work is based on GNN architecture. The entire architecture remains the same. The update of the node information is computation during message passing step which is the equation (5).


## Questions for the Authors

1. The aggregation strategy is to use node information, label information and structural information together to train a representation for node classification. The three components can not be separated. We also compared our solution with the state-of-the-art solutions and MPL as the benchmark solution.
2. There is no label matrix. Each label indicates the class of the sample.
3. For graph data, we need to have the momentum in the mind. Each node has local information and neighboring nodes spread influence to other nodes. During graph propagation, the node information is updated by neighboring nodes from 1 hop to n hop. The iterations the model runs, the further hops it can aggregate. When the change on the node is very small, we consider the graph is converged. At that moment, we can say each node becomes stable, the local information on the node has been turned into global information, and eventually we get a global representation of the nodes. We can directly use node representation to classify nodes. We use experimental results to show that, given trained representation for the nodes, we can use the trained representation to classify nodes.
4. we put the state-of-the-art research for Homophily-heterophily issue into several categories and compared our work with the algorithms in all the categories. We used GNN as the backbone algorithm, but the backbone algorithm is not the main focus of this research.
5. Eigen centrality is a graph property. We only need to perform the computation once.

---

### Meta-Review · Area_Chair_2cjp · 2026-01-17

**Recommendation:** Reject
**Confidence:** 3

**Metareview:**

This paper challenges the homophily assumption in graph data aggregation by showing that aggregation performance depends on the strategy rather than the network type. It proposes a Graph Weighted Aggregation (GWA) method that jointly leverages node features, network topology, and label information. By modeling graph topology on a Riemannian manifold and using Ricci curvature to capture inter-node influence, GWA performs principled message passing until reaching a minimum-energy state. The proposed approach is homophily-agnostic and outperforms state-of-the-art methods on benchmark datasets.

There are major concerns proposed by reviewers such as lack of clarification on core theoretical claims; lack of formalizing the mathematics with a clear energy-based framework. The rebuttal has fixed some format issues and partially addressed some concerns such as why no ablation study can be provided.

---

### Decision · Program_Chairs · 2026-01-17

**Decision:**

Accept (Poster)

**Comment:**

Based on the AC’s comments and the authors’ revisions, the PC has determined that this paper can be accepted as a poster for presentation at the workshop.

The authors are expected to thoroughly revise the manuscript in accordance with the reviewers’ and AC’s comments to further improve the quality of the work.